# Timo: Towards Better Temporal Reasoning for Language Models

**Zhaochen Su**[1,*] **Jun Zhang**[1], **Tong Zhu**[1],
**Xiaoye Qu**[2], **Juntao Li**[1,†] **Min Zhang**[1], **Yu Cheng**[3]
[1]Institute of Computer Science and Technology, Soochow University, China;
[2]Shanghai AI Laboratory; [3]The Chinese University of Hong Kong
{suzhaochen0110,junzhang20030309}@gmail.com;
{ljt,minzhang}@suda.edu.cn; quxiaoye@pjlab.org.cn;
tzhu1997@outlook.com; chengyu@cse.cuhk.edu.hk

## Abstract

Reasoning about time is essential for Large Language Models (LLMs) to understand the world. Previous works focus on solving specific tasks, primarily on time-sensitive question answering. While these methods have proven effective, they cannot generalize to a wider spectrum of temporal reasoning tasks. Therefore, we propose a crucial question: Can we build a universal framework to handle a variety of temporal reasoning tasks? To that end, we systematically study 38 temporal reasoning tasks. Based on the observation that 19 tasks are directly related to mathematics, we first leverage the available mathematical dataset to set a solid foundation for temporal reasoning. However, the in-depth study indicates that focusing solely on mathematical enhancement falls short of addressing pure temporal reasoning tasks. To mitigate this limitation, we propose a simple but effective self-critic temporal optimization method to enhance the model's temporal reasoning capabilities without sacrificing general task abilities. Finally, we develop Timo, a model designed to excel in temporal reasoning at the 7B and 13B scales. Notably, Timo outperforms the counterpart LLMs by 10.0 and 7.6 in average accuracy scores and achieves the new state-of-the-art (SOTA) performance of comparable size. Extensive experiments further validate our framework's effectiveness and its generalization across diverse temporal tasks. The code is available at `https://github.com/zhaochen0110/Timo`.

## 1 Introduction

Large Language Models (LLMs) have achieved remarkable success in various reasoning tasks (Zhao et al., 2023; Chang et al., 2023; Su et al., 2024), such as mathematical, commonsense, and symbolic reasoning. Despite these advances, LLMs face significant challenges in temporal reasoning (Chen et al., 2021; Tan et al., 2023a), which is crucial in human perception. Compared to other reasoning tasks that focus solely on one specific reasoning ability, temporal reasoning is an integrated task that requires arithmetic (Zhu et al., 2023a), logic (Mishra et al., 2022a) and world knowledge (Wei et al., 2022).

Prior efforts to improve the temporal reasoning capacity of LLMs focus mainly on time-sensitive question-answering (Chen et al., 2021), and utilize methods such as step-by-step reasoning (Zhu et al., 2023b; Li et al., 2023) and ruled-based supervised fine-tuning (SFT) (Tan et al., 2023a; Yuan et al., 2023b). More recent studies expand the scope of temporal tasks to include basic temporal concepts understanding (e.g., duration), intricate temporal interpretations (e.g., relation) and computations (e.g., arithmetic) (Wang & Zhao, 2023). Due to their task-specific nature, the aforementioned methods exhibit limited generalization across the wider spectrum of temporal tasks.

---

*Work was done during the internship at Shanghai AI lab.
†Juntao Li is the Corresponding Author.

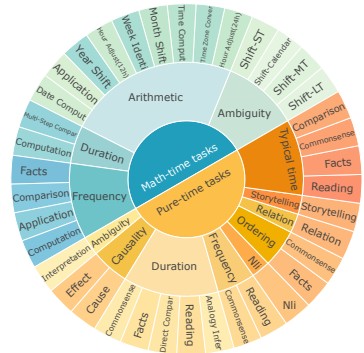 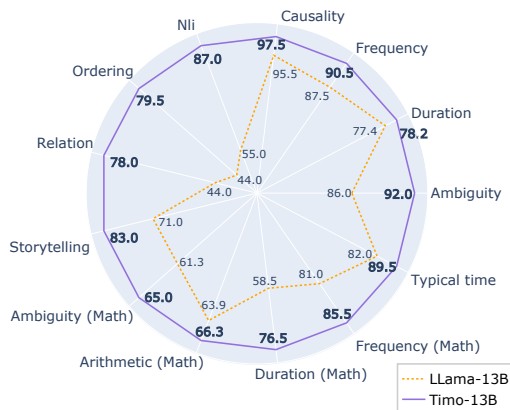

Figure 1: The detailed classification of 38 temporal reasoning tasks. 19 tasks are directly related to math (i.e., Math-time tasks).

Figure 2: Timo outperforms LLaMA in all temporal tasks and is the current state-of-the-art (SOTA) model of comparable size.

To address these limitations, we explore a crucial question: Can we build a universal framework to handle various temporal reasoning tasks? To tackle this, we face the following challenges: (1) integrating different temporal reasoning tasks into a unified framework; (2) generating and selecting the high-quality training dataset automatically; (3) improving the comprehensive temporal reasoning abilities while maintaining its general performance.

In response to these challenges, we first systematically study 38 subtasks within the temporal reasoning benchmark proposed by Wang & Zhao (2023). As shown in Figure 1, our analysis reveals that 19 tasks are directly related to mathematical reasoning (i.e., mathematical time tasks). For example, when "identifies the next leap year following 2024", mathematical skills are required to calculate the results. The rest are categorized as pure temporal reasoning tasks, focusing solely on temporal reasoning without using mathematical abilities. Meanwhile, mathematical reasoning stands out with its diverse and rich instruction tuning datasets compared to temporal reasoning (Cobbe et al., 2021; Mishra et al., 2022b; Yue et al., 2023). Therefore, it is intuitive to build a generalist temporal reasoning framework based on math-enhanced LLMs, setting a solid foundation for temporal reasoning skills. However, our in-depth study indicates that focusing solely on mathematical enhancement through supervised fine-tuning falls short of addressing pure-time tasks. To bridge this gap, we further introduce a simple but effective method to obtain comprehensive temporal reasoning abilities. Specifically, we propose a self-critic method to generate and select the high-quality temporal preference pairs, which are then utilized for enhancing model temporal capabilities through preference optimization. Finally, we propose a unified temporal reasoning framework, namely Timo. With this framework, our model achieves superior performance among 38 temporal tasks, as depicted in Figure 2.

In our experiments, we train LLaMA2 models at both 7B and 13B scales with our framework, which results in Timo-7B and Timo-13B. These two models demonstrate a substantial improvement of 10.0 and 7.6 in average accuracy scores over the base models, respectively. Our comprehensive analysis indicates that our framework successfully integrates substantial mathematical knowledge along with temporal information. Extensive experiments further verify the effectiveness of our method in preserving general task capabilities and maintaining robustness under different scenarios. To sum up, our contributions are shown below:

- We systematically study a variety of temporal reasoning tasks and discover the inner correlation between time and mathematics, where temporal reasoning could benefit from mathematics instructions;
- We make the first attempt to build a unified framework to address 38 temporal tasks. Specifically, upon mastering mathematical reasoning capabilities, we propose a simple but effective self-critic temporal optimization method to strengthen the temporal reasoning capabilities comprehensively;

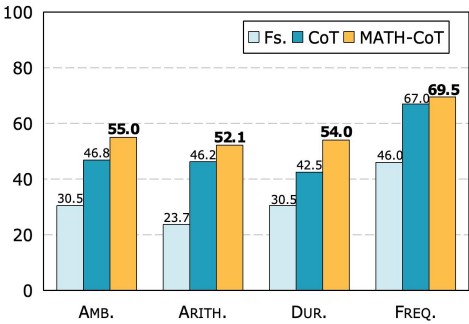

Figure 3: Performance comparison with Math-CoT and traditional prompting methods in math-time tasks.

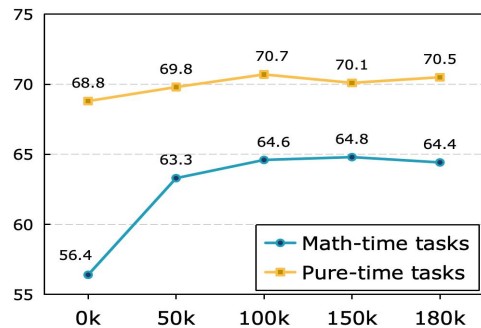

Figure 4: Comparisons on temporal tasks with models trained on different numbers of math instructions.

- The proposed framework outperforms 10.0 and 7.6 scores over the baselines, establishing as the new SOTA model of comparable sizes. Besides, our models consistently enhance the temporal reasoning capabilities without sacrificing general task performance.

## 2 Revealing the Correlation between Math and Temporal Reasoning

### 2.1 Analysis on Temporal Reasoning Benchmark

Wang & Zhao (2023) provides a comprehensive collection of 38 subtasks centered around temporal reasoning tasks. It is widely observed that a substantial portion of these tasks relies on mathematical skills for calculating and reasoning about time. For example, within the Frequency category, the Computation subtask requires the calculation of event frequencies or intervals. In the Ambiguity Resolution task, mathematics provides a standardized method of time representation, such as the 24-hour format and date calculation formulas, enabling different temporal expressions to be accurately understood and converted. Based on these observations, we categorize temporal tasks into two categories. The specific subtasks under each category are shown in Figure 1. Below is our classification:

- **Mathematical Time Tasks (Math-time tasks):** These are temporal reasoning tasks that necessitate mathematical skills, such as calculating time frequencies, converting time shifts, comparing time sequences, determining time intervals, estimating durations, and so on. This category encompasses a total of 19 subtasks.
- **Pure Time Tasks (Pure-time tasks):** These tasks require only temporal reasoning abilities for resolution and include reasoning about temporal commonsense, applications in real-world scenarios, temporal natural language inference (NLI) and so on. This category also contains 19 subtasks.

### 2.2 Bridging Mathematics and Temporal Tasks

Inspired by Wei et al. (2022), we construct Math-CoT for each temporal task to establish a connection between mathematics and temporal tasks. We utilize the MathInstruct dataset (Yue et al., 2023), which comprises a diversified collection of mathematical problems with detailed rationales. From this dataset, we select five mathematical question-CoT pairs and employ GPT-4 to generate Math-CoT rationales by mimicking mathematical reasoning. Since pure-time questions lack mathematical rationales, Math-CoT is specifically designed for math-time tasks. We compare Math-CoT with two prompting methods: (1) Few-shot, which samples five question-answer pairs per task, and (2) CoT (Wei et al., 2022), where GPT-4 is used to generate step-by-step rationales for each task. We conduct the experiments using LLaMA2-7B under the 5-shot setting and report the accuracy for each task. As shown in Figure 3, integrating mathematical reasoning into temporal tasks leads to a significant

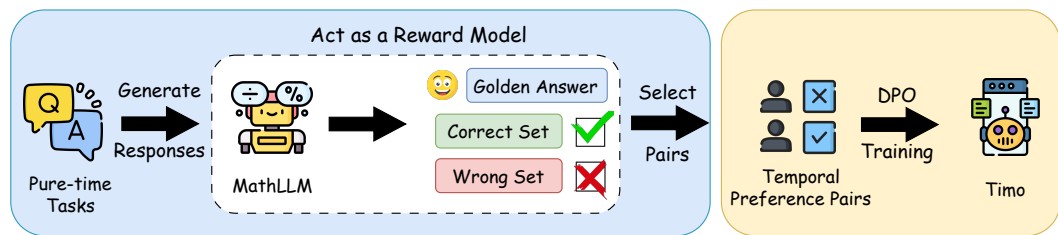

Figure 5: The pipeline of our self-critic temporal task optimization method. Based on responses generated by mathematical models (MathLLM), we classify correct and wrong sets using golden answers. From these, we further select high-quality pairs with our proposed hierarchical scoring method. Finally, the chosen pairs are used for DPO training.

enhancement in model performance, with Math-CoT outperforming traditional prompting methods in all math-time tasks.

## 2.3 Mathematical Reasoning as a Foundation for Temporal Understanding

Given the established correlation between mathematics and temporal reasoning, it is intuitive to instruct models in mastering mathematical reasoning to establish a solid foundation for advanced temporal reasoning abilities. This connection motivates our investigation into how varying degrees of mathematical instruction influence model performance. Specifically, we select 180k mathematical CoT rationales from the MathInstruct and perform scaling experiments by fine-tuning the LLaMA2-7B with different volumes of math instructions (i.e., 0, 50k, 100k, 150k, and 180k). We evaluate the models on both math-time tasks and pure-time tasks under the 5-shot setting. The results are shown in Figure 4. After supervised fine-tuning on 50k math instruction tuning instances, the model exhibits a notable improvement in performing math-time tasks, with accuracy increasing from 56.4 to 63.3. However, It is worth noting that this enhancement in mathematical skills has a minimal impact on pure-time tasks, with a maximum enhancement of 1.9. Additionally, our analysis indicates a declining trend in performance across both task categories as the volume of math instructions increases. We believe this decline results from overfitting to mathematical tasks due to excessive data, adversely impacting the model's temporal reasoning capability (Mishra et al., 2022a).

## 3 Self-critic Temporal Task Optimization

In the previous section, we discovered that focusing solely on mathematical enhancement falls short of addressing pure-time tasks. To mitigate this limitation, we introduce a simple but effective self-critic optimization framework to equip the model with comprehensive temporal reasoning abilities. The pipeline of our proposed framework is detailed in Figure 5.

Given the mathematical model $L$, we start by generating a set of $N$ candidate responses $\mathcal{Y}_i = \{y_i^1, y_i^2, \ldots, y_i^N\}$ for each input prompt $x_i$. Given the golden label $g_i$ for each prompt $x_i$, we divide $\mathcal{Y}_i$ into the correct response set $\mathbf{R}_i^+$ and the incorrect response set $\mathbf{R}_i^-$:

$$\mathbf{R}_i^+ = \{y_i^n \in \mathcal{Y}_i \mid \text{align}(y_i^n, g_i) = \text{true}\}, \quad \mathbf{R}_i^- = \mathcal{Y}_i \setminus \mathbf{R}_i^+, \tag{1}$$

where $\text{align}(y_i^n, g_i)$ is a function that returns true if the response $y_i^n$ aligns with the golden label $g_i$, and false otherwise. Inspired by the LLM-as-a-Judge mechanism (Zheng et al., 2023; Yuan et al., 2024; Qu et al., 2024b), we utilize mathematical model $L$ directly as a reward model to identify high-quality response pairs. Notably, we introduce a novel hierarchical scoring method, which is specifically designed for evaluating responses to temporal tasks and contains five key aspects: (1) relevance and basic temporal reasoning; (2) understanding of temporal aspects; (3) application of internal temporal knowledge; (4) direct and well-organized addressing of the question; (5) insightfulness and advanced reasoning. To choose the higher quality pair from the correct set $\mathbf{R}_i^+$ and wrong set $\mathbf{R}_i^-$, we

prioritize the response that utilizes the model's temporal reasoning to the fullest extent. The criteria for our evaluation prompts are illustrated in Figure 10 and 11. For each criterion a response meets, a point is awarded. We prompt the model $L$ to assign a score $r_i^n \in [0, 5]$ to each response $y_i^n$, quantifying its quality across the above dimensions.

The temporal preference pair $(y_i^+, y_i^-)$ is formed by selecting the top-scoring response from the correct set $\mathbf{R}_i^+$ as $y_i^+$ and from the incorrect set $\mathbf{R}_i^-$ as $y_i^-$. We then utilize these pairs to perform direct preference optimization (DPO) by optimizing the following loss function:

$$\mathcal{L}_{\text{DPO}}(\pi_\theta; \pi_{\text{ref}}) = -\mathbb{E}_{(x, y_i^+, y_i^-) \sim \mathcal{D}} \left[ \log \sigma \left( \beta \log \frac{\pi_\theta(y_i^+ \mid x)}{\pi_{\text{ref}}(y_i^+ \mid x)} - \beta \log \frac{\pi_\theta(y_i^- \mid x)}{\pi_{\text{ref}}(y_i^- \mid x)} \right) \right], \quad (2)$$

where $y_i^+$ is favored over $y_i^-$, and $\beta$ is a hyperparameter.

# 4 Experiments

## 4.1 Experimental Setup

**Training Setup**   We use LLaMA2 7B and 13B (Touvron et al., 2023) as our base pre-trained model. For SFT, we select 100k instances from MathInstruct (Yue et al., 2023), the most representative dataset for mathematical reasoning instruction tuning. For self-critic temporal optimization, we focus on pure temporal reasoning tasks, which encompass 19 subtasks. We reserve 100 instances for evaluation and utilize the remaining data for training. If a subtask contains fewer than 5,000 samples, we maintain all of them. Otherwise, we randomly select 5,000 instances. In total, we use 35,655 instances for optimization.

**Evaluation Setup**   We conduct a comprehensive evaluation across all temporal reasoning tasks, encompassing a total of 38 tasks. Following Tan et al. (2023a), we assess the model performance on 100 examples for each task, amounting to a total of 3,800 instances. Consistent with prior work (Qu et al., 2024a; Xia et al., 2024), we evaluate the model's temporal abilities under the 5-shot setting and utilize greedy decoding (i.e., temperature = 0) for generating model's responses. We extract the prediction from the response and calculate the accuracy of each subtask.

**Implementation Details**   We utilize four/eight NVIDIA Tesla A100 GPUs to train models. To facilitate parallel training, we employ DeepSpeed Zero-Stage 3 (Ren et al., 2021) and FlashAttention2 (Dao, 2023). For SFT, we use a learning rate of 2e-5, a batch size of 128, and a cosine scheduler with a 3% warm-up period for 2 epochs. For candidate response generation, we sample $N = 5$ candidate responses with temperature $T = 0.8$, $p = 0.95$. When evaluating candidate responses, as there is variance to these scores, in our experiments we also use sampled decoding (with the same parameters) and generate these evaluations multiple (3) times and take the average. For DPO, we follow the hyper-parameters from Tunstall et al. (2023) with a batch size 32, learning rate 5e-7, a warm ratio of 0.1 using linear warmup scheduler for 9 epochs.

## 4.2 Baselines

To ensure the fairness of the experiments, we select baseline models built upon the foundational model **LLaMA2**. The baselines are selected based on the following dimensions:

- **LLMs for Temporal Reasoning: TimeLLaMA** (Yuan et al., 2023b) is currently the only open-source model that is specifically designed for temporal reasoning. It is developed to make temporal predictions and generate time-related explanations.
- **LLMs for Mathematical Reasoning:** Timo is trained through temporal optimization based on mathematical models. Here, we compare the following mainstream mathematical models: (1) **MAmmoTH** (Yue et al., 2023) is designed for general mathematics problem-solving and is trained on the MathInstruct dataset. (2) **WizardMath** (Luo et al., 2023a)

| Model | Amb. | Ari. | Dur. | Fre. | Cau. | NLI | Ord. | Rel. | Sto. | Typ. |
|---|---|---|---|---|---|---|---|---|---|---|
| **7B Parameter Model** | | | | | | | | | | |
| LLaMA2 | 61.0 | 52.1 | 64.1 | 72.0 | 93.0 | 44.0 | 43.0 | 54.0 | 66.0 | 72.5 |
| TimeLLaMA | 56.2 | 42.7 | 37.0 | 40.8 | 42.5 | 15.0 | 7.5 | 48.0 | 5.0 | 32.0 |
| WizardCoder | 57.8 | 51.9 | 63.4 | 68.7 | 86.0 | 43.0 | 38.0 | 45.0 | 54.0 | 65.8 |
| CodeLlama | 50.2 | 55.2 | 58.7 | 71.0 | 86.0 | 55.0 | 44.0 | 45.0 | 55.0 | 68.3 |
| WizardMath | 65.2 | 52.9 | 53.3 | 72.0 | 94.0 | 49.0 | 39.0 | 36.0 | 63.0 | 64.3 |
| ToRA | 51.2 | 44.8 | 58.1 | 71.5 | 92.0 | 48.0 | 39.5 | 46.0 | 68.0 | 73.3 |
| MAmmoTH | 63.0 | 52.3 | 60.3 | 62.8 | 90.5 | 39.0 | 43.0 | 51.0 | 69.0 | 67.3 |
| Timo | **68.8** | **60.8** | **72.1** | **78.0** | **95.0** | **74.0** | **71.5** | **70.0** | **87.0** | **83.8** |
| **13B Parameter Model** | | | | | | | | | | |
| LLaMA2 | 66.2 | 63.9 | 72.0 | 83.2 | 95.5 | 55.0 | 44.0 | 44.0 | 71.0 | 82.0 |
| WizardCoder | 63.2 | 60.1 | 62.1 | 76.5 | 92.0 | 54.0 | 49.0 | 51.0 | 59.0 | 71.3 |
| CodeLlama | 62.6 | 60.6 | 67.0 | 75.2 | 90.5 | 54.0 | 46.5 | 53.0 | 58.0 | 69.5 |
| WizardMath | 63.2 | 58.3 | 72.9 | 78.3 | 95.0 | 58.0 | 43.0 | 54.0 | 82.0 | 77.3 |
| ToRA | 61.2 | 50.8 | 67.9 | 77.4 | **97.5** | 56.0 | 38.5 | 64.0 | 80.0 | 79.5 |
| MAmmoTH | 69.1 | **67.0** | 72.4 | 80.4 | 97.0 | 62.0 | 48.0 | 57.0 | 77.0 | 78.8 |
| Timo | **70.4** | 66.3 | **77.7** | **87.2** | **97.5** | **87.0** | **79.5** | **78.0** | **83.0** | **89.5** |

Table 1: Results on 38 temporal reasoning tasks. The abbreviations Amb., Ari., Dur., Fre., Cau., Ord., Rel., Sto., Typ. refer to ambiguity resolution, arithmetic, duration, frequency, causality, order, relation, story, and typical time. All values are percentages. The best results are in **bold** and the second results are in underlined.

utilizes the proposed Reinforcement Learning from Evol-Instruct Feedback (RLEIF) (Xu et al., 2023) to enhance its mathematical reasoning capabilities. (3) **ToRA** (Gou et al., 2024), a series of Tool-integrated Reasoning LLM Agents, is designed to solve challenging mathematical reasoning problems.

- **LLMs for Code Generation:** Previous work indicates that the usage of code enhances the model's ability to solve reasoning tasks (Gao et al., 2023). We select the following popular code models as our baselines: (1) **CodeLLaMA** (Roziere et al., 2023), a family of LLMs for code generation and programming-related tasks. (2) **WizardCoder** (Luo et al., 2023b) is similar to WizardMath and adapts the RLEIF method within the domain of coding.

### 4.3 Main Results

Table 1 presents the results of Timo across 38 temporal reasoning tasks. Results categorized by Math-time and Pure-time tasks are shown in Appendix G in Table 9 and Table 10. From the results, we observe: (1) Timo surpasses counterpart LLMs in average accuracy of 10.0 and 7.6 scores, and outperforms other competitive math-solving and code-solving models with a clear margin, achieving the SOTA results on average. We also discover that Timo-7B consistently outperforms all 13B models in average performance, achieving a maximum performance gain of 7.1. (2) Mathematical models do not show significant advantages in solving math-related tasks. This phenomenon is also observed in LLMs enhanced for coding abilities and temporal prediction capabilities. It indicates that excessive training on specific abilities leads the model to overfit in task-centric enhancements, diminishing its performance in other areas (Jha et al., 2023). (3) It is worth noting that Timo underperforms MAmmoTH in the Arithmetic task (i.e., scoring 66.3 vs 67.0) when evaluated under the 13B model size parameter. The superior performance of MAmmoTH can be attributed to its advanced general math-solving abilities, which facilitate more accurate computations in time-related scenarios. However, other mathematical models like ToRA and WizardMath do not achieve the same effectiveness in handling the Arithmetic task. A detailed case study for illustration is in Appendix B.

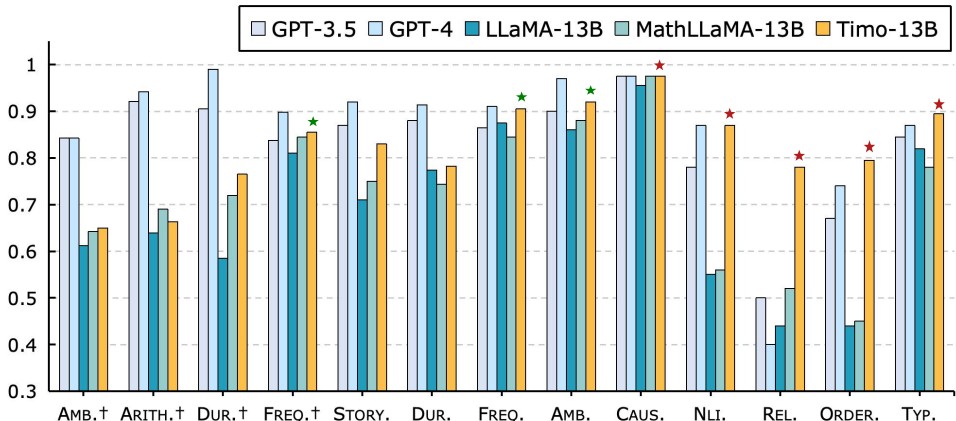

Figure 6: Performance of GPT series and our framework's models. MathLLaMA is based on mathematical instruction tuning and Timo is our final model. Math-time tasks are marked with †, while others are pure-time tasks. We highlight our model's achievements: a green star (⋆) where Timo beats GPT-3.5, and a red star (⋆) for surpassing GPT-4.

## 5 Further Analysis on our Framework

In our framework, we initially train a mathematical model, i.e., MathLLaMA. Then, we optimize its pure temporal reasoning abilities to derive the final Timo model. In this section, we first compare the performance of these two stages. Then, we delve into these models through the lens of token distribution shift and detailed case analysis.

**Ablation Analysis of Framework**   We compare the model's performance on both math-time tasks and pure-time tasks. The results are shown in Figure 6. Compared to the foundational model LLaMA, MathLLaMA demonstrates superior performance in math-related tasks and surpasses the LLaMA in the majority of pure-time tasks, achieving higher scores in 6 out of 9 tasks. This improvement is attributed to the advanced logical and reasoning skills developed through mathematical instruction tuning (Mishra et al., 2022a). When compared to Timo and MathLLaMA, our framework demonstrates strong generalization capabilities, achieving significant improvement in pure-time tasks, with only minimal performance degradation in the arithmetic task. Additionally, it is worth noting that Timo outperforms MathLLaMA in various math-time tasks (i.e., Ambiguity Resolution, Duration and Frequency). This improvement is attributed to our framework's ability to learn generalized temporal features.

**Token Distribution Shift Analysis**   To understand the learning process and the differences between the different stages of our framework, we follow the methodology proposed by Lin et al. (2024) to analyze through the lens of token distribution shift. We analyze three pairs of models at the 7B scale: LLaMA vs MathLLaMA, MathLLaMA vs Timo, and LLaMA vs Timo. The results are shown in Figure 7. Notably, we observe the largest token distribution shift when transitioning from LLaMA to Timo. Furthermore, we investigate the top 200 most frequently shifted tokens, labeling math-related tokens in red and time-related tokens in green. The transition from LLaMA to MathLLaMA primarily features changes in math-related tokens. Conversely, the switch from MathLLaMA to Timo is characterized by the presence of time-related tokens. When compared to LLaMA, Timo exhibits shifts in both math-related and time-related tokens, demonstrating a profound capacity to integrate substantial mathematical knowledge along with the temporal information.

**Case Analysis**   As shown in Table 2, we present a case analysis to provide a clear and intuitive demonstration of Timo's superior performance. In math-time tasks, both MathLLaMA and Timo effectively integrate temporal knowledge with computational capabilities to give the correct CoT and answer. However, LLaMA produces an incorrect result due to the

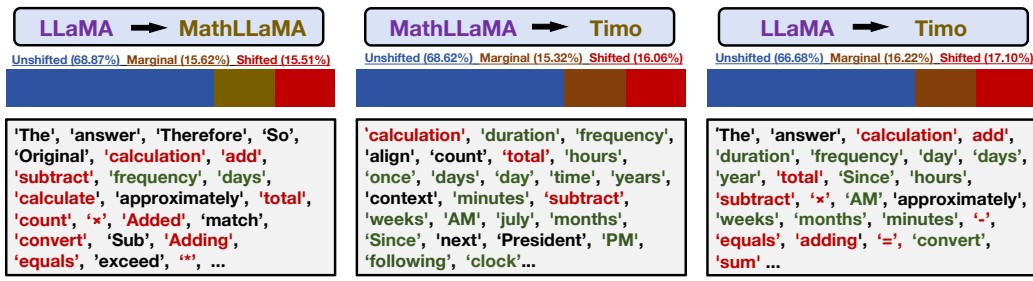

Figure 7: Token distribution shift on different stages of our proposed temporal task optimization method. The ratios of unshifted, marginal, and shifted tokens are colored (%). Frequently shifted tokens are shown below, where math-related tokens are labeled in red and time-related tokens are highlighted in green.

error in time calculation, which indicates the importance of mathematical skills in solving math-time tasks. We also In our provided case of the pure-time tasks, both MathLLaMA and LLaMA fail to grasp the sequence of events, i.e., the timing of Amy's laundry activities. On the other hand, Timo demonstrates a strong understanding and application of temporal reasoning, accurately tracking the sequence and timing of Amy's activities and giving the correct answer. Overall, these cases vividly demonstrate Timo's comprehensive capabilities in temporal reasoning across different temporal task types.

## 6   More Detailed Study

**Performance Comparison between Timo and OpenAI GPT Models**   We compare Timo-13B with the current most powerful LLMs, i.e., GPT-3.5 and GPT-4. Specifically, we use the gpt-3.5-turbo-1106 and gpt-4-1106-preview and set the temperature to 0 for consistent evaluation. The results are shown in Figure 6. Despite its relatively small size of 13B parameters, Timo demonstrates impressive performance on pure-time tasks, outperforming GPT-3.5 in 7 out of 9 tasks and surpassing GPT-4 in 5 out of 9 tasks. Notably, Timo exceeds GPT-4 by a significant margin of 38 accuracy scores in the Relation task. Although there has been a significant improvement in pure-time tasks, the performance on math-time tasks suggests that there is still room for further enhancement. This is attributed to the foundational model's weaker mathematical reasoning capabilities. We leave it as future work to further improve the model's temporal reasoning abilities by better integrating mathematics capabilities.

**Performance Comparison among Different Rewarding Strategies**   In our framework, we design a series of criteria to assess the standard of responses and obtain high-quality temporal preference pairs. To verify the effectiveness of our criteria, we compare our prompting approach with the widely adopted self-rewarding strategy (Yuan

|           | Math-time | Pure-time |
|-----------|-----------|-----------|
| random    | 61.5      | 79.8      |
| LLM-Judge | 61.3      | 80.2      |
| Timo      | **63.9**  | **81.5**  |

Table 3: Performance on different reward methods

et al., 2024) and the random selection strategy. As shown in Table 3, our strategy outperforms others in both math-time and pure-time tasks, highlighting its superiority in evaluating the quality of generated responses across different types of temporal challenges.

**Robustness across Mathematical Models**   With Timo being derived from a mathematical model trained with 100k math instructions, we validate the robustness and adaptability of our framework across different mathematical models, which is achieved by implementing self-critic temporal task optimization in models supervised fine-tuned with different volumes of instruction dataset (i.e., 50k, 100k, 150k, 180k). The results are shown in Fig-

| Mathematical Time Tasks | |
|---|---|
| **Questions** | In which week of year 1652 does the date 10-27-1652 occur? 
 (A) Week 43; (B) Week 39; (C) Week 46; (D) Week 30 |
| **LLaMA** | Calculating the weeks from January 1st to October 27th, it's 27 days. Dividing 27 by 7 places October 27th in the 4th week. 
 However, depending on the week starting day and system used, it can be considered as the 39th week. The answer is (B). ✗ |
| **MathLLaMA** | We calculate the total days from January 1st to October 27th, which amounts to 31 (Jan.) + 28 (Feb.) + 31 (Mar.) + 30 (Apr.) + 31 (May.) + 30 (Jun.) + 31 (Jul.) + 31 (Aug.) + 30 (Sept.) + 27 (Oct.), which equals 300 days. Dividing 300 by 7, we find that October 27th falls into the 43rd week of the year. The answer is (A). ✓ |
| **Timo** | Calculating the weeks from January 1st to October 27th, the total days are 31 (Jan.) + 28 (Feb.) + 31 (Mar.) + 30 (Apr.) + 31 (May.) + 30 (Jun.) + 31 (Jul.) + 31 (Aug.) + 30 (Sept.) + 27 (Oct.), summing up to 300 days. Dividing 300 by 7 places October 27th in the 43rd week of the year. The answer is (A). ✓ |

Table 2: A case study for math-time tasks to intuitively show the effectiveness of Timo with temporal direct preference optimization. The right and wrong steps are colored by blue and red, respectively. An example of pure-time tasks is shown in Table 11.

ure 8. The experimental results indicate that the trained models consistently outperform in handling time-related tasks compared to their corresponding mathematical models, highlighting our method's capability to enhance temporal reasoning across different mathematical training backgrounds.

**General Tasks Capability Assessment** To verify the model's ability to retain its original capabilities, we utilize the lm-evaluation-harness (Gao et al., 2021) to evaluate its performance on six typical downstream tasks: 5-shot MMLU (Hendrycks et al., 2020), 25-shot ARC Challenge (Clark et al., 2018), 5-shot GSM8K (Cobbe et al., 2021), 10-shot HellaSwag (Zellers et al., 2019), 5-shot Winogrande (Sakaguchi et al., 2021) and 0-shot TruthfulQA (Lin et al., 2022). In addition to comparing with LLaMA and MathLLaMA, we introduce Timo-SFT, which mirrors our framework in all aspects except for its training methodology. Specifically, Timo-SFT is supervised fine-tuned with the chosen responses in the selected preference pairs. The results are shown in Figure 9. We surprisingly discover that Timo outperforms other baselines in the reasoning and general knowledge ability tasks. Error analysis shows that our model aligns with the base model for 97% of the correct responses. This consistency indicates that our Timo effectively preserves general task knowledge, demonstrating remarkable generalization capabilities.

# 7 Related Work

**Temporal Reasoning in LLMs** Time is a crucial dimension in our physical world (Lazaridou et al., 2021; Su et al., 2022; 2023; Zhao et al., 2024). Despite the advanced capabilities of LLMs in various tasks, their reasoning abilities are still underdeveloped (Su et al., 2024; Qiao et al., 2023; Huang & Chang, 2023; Sun et al.; Chu et al., 2023). Temporal reasoning, which is fundamental for humans to understand the world, is an important task in reasoning and has gained substantial research focus (Pustejovsky, 2003; UzZaman et al., 2012; Huang et al., 2024). However, existing works often specialize in limited aspects of temporal reasoning, such as frequency (Zhou et al., 2019), duration (Zhang & Choi, 2021), or event-time relations (Chen et al., 2021; Tan et al., 2023a). In our work, we address a comprehensive scope of temporal reasoning, including various levels and dimensions of understanding about time (Wang & Zhao, 2023). Differing from prior approaches that rely on external knowledge (Yuan et al., 2023a; Tan et al., 2023b; Xiong et al., 2024) or impose temporal constraints (Li et al., 2023; Zhu et al., 2023b) within a narrow sub-scope of tasks, we propose a unified framework designed to generalize across different temporal reasoning scenarios.

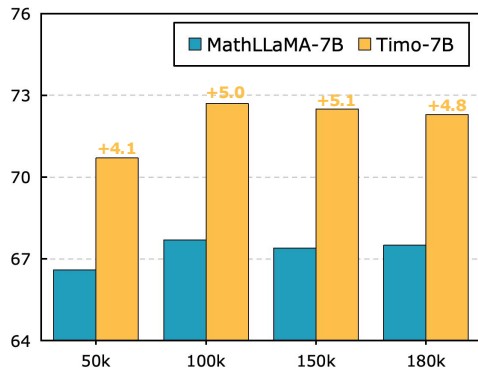

Figure 8: Results of Timo trained on the mathematics dataset of different sizes, demonstrating consistent improvements across models.

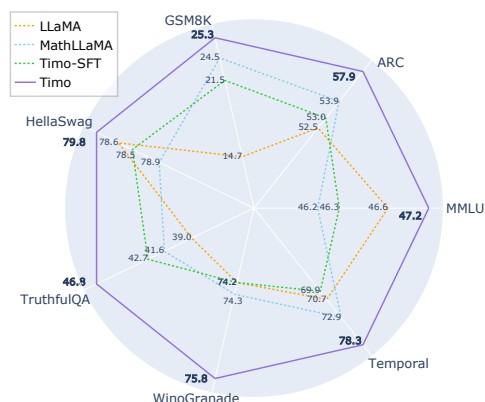

Figure 9: Reasoning and general knowledge performance comparison under current mainstream benchmarks.

**Preference Optimization for LLMs** Preference optimization approaches typically involve training a fixed reward model based on preference data, and then utilizing the reward model to train via reinforcement learning (RL) (Schulman et al., 2017; Ziegler et al., 2019; Stiennon et al., 2020; Bai et al., 2022). To simplify this process, Direct Preference Optimization (DPO) (Rafailov et al., 2023) is introduced to avoid training the reward model entirely, and instead directly train the LLM using preference pairs. Building on this approach, recent works explore automatic optimization and self-correction in LLMs (Pan et al., 2023; Ji et al., 2024). This involves two key steps: instructing LLMs to self-generate their training dataset (Wang et al., 2023; Taori et al., 2023; Tunstall et al., 2023) and serving LLMs as reward models (Fernandes et al., 2023; Saha et al., 2023; Dubois et al., 2024) to select high-quality data. The self-generated data optimization enables models to iteratively improve their performance through a self-rewarding mechanism (Yuan et al., 2024). Inspired by the above works, we introduce a self-critic temporal optimization method that leverages the inherent capabilities of the model itself to achieve significant improvements in all temporal tasks.

## 8 Conclusion

In this paper, we consider the problem of building a universal framework to strengthen the temporal reasoning capabilities of LLMs. Through systematic investigation, we discover a close relationship between mathematics and temporal reasoning. Building upon this insight, we introduce a self-critic temporal optimization method to equip the model with comprehensive temporal reasoning abilities. The Timo model, trained within our proposed framework, indicates significant generalizability across 38 temporal tasks, establishing as the new SOTA model of comparable sizes. Extensive experiments further demonstrate the effectiveness of our framework in maintaining general task abilities.

## Acknowledgement

We want to thank all the anonymous reviewers for their valuable comments. We are also grateful to Wei Liu for his insightful suggestions during the mathematical reasoning experiments. This work was supported by the National Science Foundation of China (NSFC No. 62206194), the Priority Academic Program Development of Jiangsu Higher Education Institutions, the Natural Science Foundation of Jiangsu Province, China (Grant No. BK20220488), and Young Elite Scientists Sponsorship Program by CAST (2023QNRC001).

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

## A    Prompt

Our rewarding prompts are shown in Figure 10 and  11. The prompts for different temporal tasks can be found in our GitHub repository: `https://github.com/zhaochen0110/Timo`.

## B    Comparative Analysis of Mathematical Models on Arithmetic Tasks

We observe that task-specific fine-tuning approaches compromise the LLMs' original abilities. We conduct a case study to delve deeper into this phenomenon. As shown in Table 4, ToRA and WizardMath have difficulty grasping basic concepts of time. Specifically, these models demonstrate challenges in accurately converting between the 12-hour and 24-hour time formats, a fundamental aspect of temporal understanding. This case study serves as a clear illustration of how specialized fine-tuning can compromise the LLMs' inherent ability to perform fundamental reasoning, underscoring the need for a balanced approach in model training.

| Questions | What is 02:45 PM + 10:44? 
 (A) 2:39 AM (B) 3:41 AM (C) 1:29 AM (D) 11:20 PM |
|---|---|
| **Timo** | The calculation is $14:45 + 10:44 = 25:29$, which is 1:29 AM. The answer is (C). |
| **MammoTH** | The calculation is $14:45 + 10:44 = 25:29$, which is 1:29 AM. The answer is (C). |
| **WizardMath** | The calculation is $15:45 + 10:44 = 26:29$, which is 3:41 AM. The answer is (B). |
| **ToRA** | The calculation is $14:45 + 10:44 = 25:29$, which is 3:41 AM. The answer is (B). |

Table 4: The case chosen from the Hour Adjustment (12h) task. ToRA and WizardMath fall short in time calculation and converting between 12-hour and 24-hour formats.

## C    Iterative Optimization Study

Recent work (Touvron et al., 2023; Yuan et al., 2024) suggests that updating preference data through multiple iterative rounds enhances the performance of preference optimization. Therefore, we explore Iterative DPO to refine alignments across temporal reasoning tasks. The results are shown in Table 5. However, we do not observe a significant improvement from iterative training. The reason might be due to the efficiency of our method, where a single iteration is sufficient for robust learning, and excessive training could instead diminish performance in temporal reasoning tasks.

|  | Math-time | Pure-time |
|---|---|---|
| **1 iter.** | **63.9** | **81.5** |
| **2 iters** | 62.1 | 80.9 |
| **3 iters** | 57.8 | 80.1 |

Table 5: Comparison on different iteration settings

## D    Validating Timo on LLaMA3-8B

To further validate the effectiveness of Timo in enhancing temporal reasoning across different LLMs, we conducted additional experiments using the LLaMA3-8B model. The results are shown in Table 6. Compared to vanilla LLaMA3-8B, Timo shows an average improvement of 5.1 scores, with 1.2 scores in math-related tasks and 9 scores in time-related tasks. These consistent improvements across both the LLaMA2 and LLaMA3 series demonstrate Timo's strong generalization capabilities across different model series, enhancing its applicability and effectiveness in diverse settings.

|              | Math-time | Pure-time | Average |
| --- | --- | --- | --- |
| **LLaMA3-8B** | 81.4 | 79.6 | 80.5 |
| **+Timo** | **82.6** | **88.6** | **85.6** |

Table 6: Performance Comparison of LLaMA3-8B with and without Timo enhancement

# E   Evaluating the Impact of Math LLM on Temporal Reasoning

Existing work on weak-to-strong generalization suggests that distilling data from a weaker or equivalent LLM benefits a stronger LLM (Burns et al., 2023). To address concerns regarding the influence of the LLM-as-Judge framework compared to the use of a specialized math LLM, we conducted experiments using vanilla LLaMA2-7B and LLaMA2-7B-chat, representing general SFT LLaMA models. As presented in Table 7, our results demonstrate that incorporating a math LLM yields significant improvements in temporal reasoning tasks. Specifically, the math LLM outperforms the vanilla LLaMA2-7B and LLaMA2-7B-chat models by an average of 3.6 and 7 scores, respectively. The performance gains are especially notable in math-related tasks, where the math LLM achieves scores 5.5 and 10.5 scores higher than those of the other two models. These results indicate that the math LLM component is crucial for enhancing temporal reasoning capabilities, outperforming the self-critic temporal optimization (i.e., LLM-as-Judge) framework alone. The results indicate that math-specific training plays a pivotal role in reasoning over time, confirming the value of specialized LLMs in complex reasoning tasks.

|                            | Math-time | Pure-time | Average |
| --- | --- | --- | --- |
| **Timo (LLaMA2-7B)** | 58.4 | 79.7 | 69.1 |
| **Timo (LLaMA2-7B-chat)** | 53.4 | 78.1 | 65.7 |
| **Timo (MathLLaMA-7B)** | 63.9 | **81.5** | **72.7** |

Table 7: Comparison of temporal reasoning performance across different based LLM, with Timo applied for temporal optimization on LLaMA2-7B, LLaMA2-7B-chat, and MathLLaMA-7B.

# F   Further Evaluation of Timo on Temporal Reasoning Datasets

To further assess Timo's improvements in temporal reasoning, we extended our evaluation to additional temporal reasoning datasets, i.e., MCTACO (Zhou et al., 2019) and TempReason (Tan et al., 2023a). These datasets were selected to validate Timo's effectiveness across a broader range of temporal reasoning tasks.

- **MCTACO**: This dataset evaluates a wide range of commonsense knowledge related to events, including the duration, order, stationary nature, and typical timing of events.

- **TempReason**: This dataset emphasizes implicit temporal reasoning in structured facts, focusing on both event-time reasoning and event-event reasoning.

The results are shown in Table 8. Timo achieves scores 6.2 and 15.5 points higher than LLaMA2-7B and WizardMath-7B on the TempReason task. Additionally, Timo surpasses MAmmoTH-7B by 19.3 points on the MCTACO task. These results indicate that Timo excels across various temporal reasoning datasets, demonstrating its robust general temporal reasoning abilities. In future work, We will further explore the generalization of Timo across more reasoning tasks, such as commonsense reasoning (Sakaguchi et al., 2021), and composition relations reasoning (Zhao & Zhang, 2024).

|              | MCTACO | TempReason |
|--------------|--------|------------|
| **LLaMA2-7B**    | 50.3   | 46.6       |
| **MAmmoTH-7B**   | 37.0   | **52.8**   |
| **WizardMath-7B**| 12.7   | 37.3       |
| **Timo-7B**      | **56.3** | **52.8** |

Table 8: Results on the MCTACO and TempReason datasets. Timo-7B outperforms its counterparts, demonstrating superior general temporal reasoning abilities.

## G   Results for Math-time and Pure-time Tasks

In this section, we present the detailed results by task category. Results for math-related tasks are presented in Table 9. Results for pure-time tasks are presented in Table 10.

| Model | Amb. | Ari. | Dur. | Fre. |
|-------|------|------|------|------|
| **7B Parameter Model** | | | | |
| LLaMA2     | 55.0 | 52.1 | 54.0 | 69.5 |
| TimeLLaMA  | 52.5 | 42.7 | 42.5 | 55.5 |
| WizardCoder| 53.8 | 51.9 | 40.5 | 66.0 |
| CodeLlama  | 44.5 | 55.2 | 50.5 | 68.0 |
| WizardMath | 63.3 | 52.9 | 45.0 | **74.0** |
| ToRA       | 45.5 | 44.8 | 44.0 | 69.8 |
| MAmmoTH    | 62.0 | 52.3 | 54.5 | 59.5 |
| Timo       | **65.3** | **60.8** | **59.5** | 72.0 |
| **13B Parameter Model** | | | | |
| LLaMA2     | 61.3 | 63.9 | 58.5 | 81.0 |
| WizardCoder| 58.5 | 60.1 | 55.5 | 72.3 |
| CodeLlama  | 57.8 | 60.6 | 61.0 | 74.8 |
| WizardMath | 58.8 | 58.3 | 62.0 | 75.5 |
| ToRA       | 56.8 | 50.8 | 48.0 | 75.8 |
| MAmmoTH    | 64.9 | **67.0** | 71.0 | 79.8 |
| Timo       | **65.0** | 66.3 | **76.5** | **85.5** |

Table 9: Results on Math-time tasks. Best results are in **bold** and the second results are underlined.

| Model | Amb. | Dur. | Fre. | Cau. | NLI | Ord. | Rel. | Sto. | Typ. |
|---|---|---|---|---|---|---|---|---|---|
| **7B Parameter Model** | | | | | | | | | |
| LLaMA2 | **85.0** | 68.2 | 77.0 | 93.0 | 44.0 | 43.0 | 54.0 | 66.0 | 72.5 |
| TimeLLaMA | 71.0 | 34.8 | 11.5 | 42.5 | 15.0 | 7.5 | 48.0 | 5.0 | 32.0 |
| WizardCoder | 74.0 | 58.6 | 74.0 | 86.0 | 43.0 | 38.0 | 45.0 | 54.0 | 65.8 |
| CodeLlama | 73.0 | 62.0 | 77.0 | 86.0 | 55.0 | 44.0 | 45.0 | 55.0 | 68.3 |
| WizardMath | 73.0 | 56.6 | 68.0 | 94.0 | 49.0 | 39.0 | 36.0 | 63.0 | 64.3 |
| ToRA | 74.0 | 63.8 | 75.0 | 92.0 | 48.0 | 39.5 | 46.0 | 68.0 | 73.3 |
| MAmmoTH | 67.0 | 62.6 | 69.5 | 90.5 | 39.0 | 43.0 | 51.0 | 69.0 | 67.3 |
| Timo | 83.0 | **77.2** | **90.0** | **95.0** | **74.0** | **71.5** | **70.0** | **87.0** | **83.8** |
| **13B Parameter Model** | | | | | | | | | |
| LLaMA2 | 86.0 | 77.4 | 87.5 | 95.5 | 55.0 | 44.0 | 44.0 | 71.0 | 82.0 |
| WizardCoder | 82.0 | 69.0 | 85.0 | 92.0 | 54.0 | 49.0 | 51.0 | 59.0 | 71.3 |
| CodeLlama | 82.0 | 69.4 | 76.0 | 90.5 | 54.0 | 46.5 | 53.0 | 58.0 | 69.5 |
| WizardMath | 81.0 | 77.2 | 84.0 | 95.0 | 58.0 | 43.0 | 54.0 | 82.0 | 77.3 |
| ToRA | 79.0 | 75.8 | 80.5 | **97.5** | 56.0 | 38.5 | 64.0 | 80.0 | 79.5 |
| MAmmoTH | 86.0 | 73.0 | 81.5 | 97.0 | 62.0 | 48.0 | 57.0 | 77.0 | 78.8 |
| Timo | **92.0** | **78.2** | **90.5** | **97.5** | **87.0** | **79.5** | **78.0** | **83.0** | **89.5** |

Table 10: Results on Pure-time tasks. All values are percentages. Best results are in **bold** and second best are underlined.

| | **Pure Time Tasks** |
|---|---|
| **Questions** | According to the Premise: Amy started doing her laundry early in the morning every weekend. answer the question: What's the more plausible CAUSE? (A) It allowed her to have the washing machines to herself as most people do laundry later in the day. (B) Amy found more lost socks in the laundry room. |
| **LLaMA** | The more plausible cause is that she found more lost socks in the laundry room. This explanation aligns with the purpose of laundry, which is to clean and organize clothes, and is a logical response to Amy's need to find lost socks. The answer is (B). ✗ |
| **MathLLaMA** | The more plausible cause is that she found more lost socks in the laundry room. This explanation aligns with the purpose of laundry, which is to clean and organize clothes, and the additional socks found in the laundry room would be a logical outcome. The answer is (B). ✗ |
| **Timo** | The reason is it allowed her to have the washing machines to herself as most people do laundry later in the day. This explanation aligns with the practicality of early morning laundry, as it minimizes the likelihood of encountering other people using the laundry facilities. The answer is (A). ✓ |

Table 11: A case study for pure-time tasks to intuitively show the effectiveness of Timo with temporal direct preference optimization. The right and wrong steps are colored by blue and red, respectively.

Use the following 5-point scoring system to review the user's question and the corresponding correct answer:

- Add 1 point if the response is ***relevant to the user's inquiry and shows some temporal reasoning ability***, even if it is incomplete or contains irrelevant content.
- Add another point if the ***response correctly understands the temporal aspects of the user's question*** but doesn't fully resolve the query or provide a direct answer.
- Award a third point if the response ***accurately uses internal temporal knowledge to effectively answer the basic elements of the user's question***.
- Grant a fourth point if the response not only effectively uses temporal reasoning but is also clearly written from an AI Assistant's perspective, ***addressing the user's question directly and comprehensively, and is well-organized***.
- Bestow a fifth point for a response that excellently applies temporal reasoning, ***reflecting expert knowledge in time-based queries, and demonstrates a high-quality, engaging, and insightful answer***.

User: **{ prompt }**
<response> **{ response }** </response>

After examining the user's instruction and the response:
- Conclude with the score using the format: "Score: **<total points>**".

Figure 10: The prompt for our LLM to act as a chosen reward model

Use the following 5-point scoring system to review the user's question and the corresponding incorrect answer that attempts to use temporal knowledge but fails to correctly solve the problem:

- Add 1 point if the response attempts to be ***relevant to the user's inquiry and shows an attempt at temporal reasoning***, even if the information is incorrect.
- Add another point if the response partially ***correctly attempts to handle the temporal aspects of the user's question but includes errors or misconceptions***.
- Award a third point if the response clearly tries to ***use temporal knowledge but fails to accurately address the basic elements of the user's question***.
- Grant a fourth point if the response, while attempting to reason temporally from an AI Assistant's perspective, ***shows some logic but contains errors or misses key information***.
- Bestow a fifth point for those answers that ***make an effort in temporal reasoning but are incorrect, potentially misleading the user but not completely deviating from the topic of the question***.

User: **{ prompt }**
<response> **{ response }** </response>

After examining the user's instruction and the response:
- Conclude with the score using the format: "Score: **<total points>**".

Figure 11: The prompt for our LLM to act as a rejected reward model

