# OpenReview forum: "Timo: Towards Better Temporal Reasoning for Language Models"
_colmweb.org/COLM/2024/Conference — COLM_

### Official Review · Reviewer_P1tK · 2024-05-03

**Rating:** 5
**Confidence:** 4
**Ethics Flag:** 1

**Summary:**

This paper presents a framework to improve the general temporal reasoning capabilities of LLMs. The authors empirically verify the intuition that temporal reasoning is closely related to math reasoning, which motivates the proposed approach that adopts the LLM-as-Judge scheme using a math LLM and temporal reasoning instructions to produce temporal-related pairwise preference dataset, which is later adopted for training a DPO model called TIMO. The authors conduct experiments on TRAM (a temporal reasoning benchmark consisting of 10 categories) and show that TIMO outperforms the math LLM under most cases.

**Reasons To Accept:**

1. This work empirically shows that temporal reasoning benefits from better math reasoning abilities. Such an assumption has been indirectly studied before, but no prior work has really proved it.

2. The produced TIMO checkpoint, upon being publicly released, could be a good option for direct usage on temporal reasoning tasks.

**Reasons To Reject:**

1. The final DPO step uses the data distilled and evaluated by the math-LLM through the LLM-as-Judge framework. However, there is no comparison with adopting a non-math LLM (e.g., vanilla LLaMA or generally-SFT LLaMA) for generating such data. Existing work on weak-to-strong generalization suggests that distilling data from a weaker or equivalent LLM benefits a stronger LLM. Thus, it is possible that the improvements shown by TIMO results from the LLM-as-Judge framework rather than the math LLM (which is considered as the main contribution of this work).

2. Among all models that are compared against TIMO, TIMO is the only one that has been trained on the instructions from TRAM. So one concern is whether the improvements of TIMO over math-LLM result from better general temporal reasoning or specialized adaptation on TRAM. Some evaluations on other held-out temporal reasoning datasets could be much more convincing.

---

> ### Author Rebuttal · Authors · 2024-05-31
>
> Dear Reviewer P1tK,
>
> Thanks for your feedback, which is crucial in improving this work.
>
> We hope the below experimental results and explanation can mitigate your concerns and answer your questions.
>
> > Concern 1: Lack of comparison with non-math LLMs (e.g., vanilla LLaMA or general SFT LLaMA). The improvements shown by TIMO might result from the LLM-as-Judge framework rather than the math LLM.
>
> Thanks for your suggestions!
>
> We conduct experiments on vanilla LLaMA-7B and LLaMA2-7B-chat (as a general SFT LLaMA) to validate the importance of math LLM in our framework.
>
> As shown in the table below, utilizing the math LLM outperforms vanilla LLaMA-7B and LLaMA2-7B-chat with average improvements of **3.6 and 7 scores**, respectively.
>
> Specifically, in math-related tasks, our model achieves 5.5 and 10.5 scores higher than the other two models.
>
> The above results indicate that using a math LLM plays a more important role in reasoning over time compared to self-critic temporal optimization (LLM-as-Judge framework).
>
> | | Math-related task | Time-related task | Average  |
> | ----| -| ---| ----|
> | TIMO (LLaMA2-7B)      | 58.4              | 79.7              | 69.1     |
> | TIMO (LLaMA2-7B-chat) | 53.4              | 78.1              | 65.7     |
> | TIMO (MathLLaMA-7B)   | **63.9**      | **81.5**          | **72.7** |
>
> > Concern 2: Evaluations on other temporal reasoning datasets could be more convincing.
>
>
> Following your comments, we evaluate TIMO on two temporal reasoning datasets to validate the effectiveness, i.e., MCTACO [1] and TempReason [2].
>
> As shown in the table below, TIMO outperforms LLaMA2-7B and WizardMath-7B by **6.2 and 15.5 scores** on the TempReason task, respectively, and exceeds MAmmoTH-7B by **19.3 scores** on the MCTACO task.
>
> These results indicate that the improvements of TIMO over math-LLM result from better general temporal reasoning rather than specialized adaptation to TRAM.
>
> |  | MCTACO   | TempReason |
> | -| -| --|
> | LLaMA2-7B     | 50.3     | 46.6       |
> | MAmmoTH-7B    | 37.0     | **52.8**   |
> | WizardMath-7B | 12.7     | 37.3       |
> | TIMO-7B       | **56.3** | **52.8**   |
>
> [1] Towards Benchmarking and Improving the Temporal Reasoning Capability of Large Language Models, ACL-23
>
> [2] “Going on a vacation” takes longer than “Going for a walk”: A Study of Temporal Commonsense Understanding, EMNLP-19

---

> > ### Author Response · Authors · 2024-06-04
> > **Looking forward to further discussion**
> >
> > Dear Reviewer P1tK,
> >
> > To address your concerns, we have conducted experiments on vanilla LLaMA-7B and LLaMA2-7B-chat to validate the importance of the math LLM in our framework. Additionally, we have evaluated TIMO on MCTACO and TempReason to demonstrate its effectiveness.
> >
> > Since the rebuttal period is closing very soon, could you please check our response to see whether it mitigates your concerns?
> >
> > Feel free to reach out with any further questions or suggestions!
> >
> > Best,
> >
> > Authors of TIMO

---

> > > ### Comment · Reviewer_P1tK · 2024-06-05
> > > **reply to rebuttal**
> > >
> > > Thank the authors for providing more information and results of the proposed model.
> > > I agree that the new information and results can address some of the issues.
> > > I will raise my score.

---

### Official Review · Reviewer_QQVe · 2024-05-11

**Rating:** 7
**Confidence:** 4
**Ethics Flag:** 1

**Summary:**

Comments after Author Discussion
Reasoning about time is essential for Large Language Models (LLMs) to understand the world. In this paper, the authors propose a simple but effective self-critic temporal optimization method (TIMO) to enhance the model’s temporal reasoning capabilities without sacrificing general task abilities. It’s designed to excel in temporal reasoning at the 7B and 13B scales and outperforms the LLaMA2 by 10 and 7.6 in average accuracy scores on 38 Temporal reasoning tasks.

**Questions To Authors:**

None

**Reasons To Accept:**

1. In this paper, the authors discover the inner correlation between time and mathematics, where temporal reasoning could benefit from math instructions by systematically studying diverse temporal reasoning tasks.

2. The authors make the first attempt to build a unified framework to address 38 temporal tasks. They propose a simple but effective self-critic temporal optimization method to strengthen the temporal reasoning capabilities comprehensively upon mastering mathematical reasoning capabilities.

**Reasons To Reject:**

1.In section ‘Main Results’, there is lack of analysis on comparison with the other models.
2. In section ‘Ablation Analysis of Framework’, there is lack of analysis on comparison with the GPT4 model. Why do the performance differences between TIMO and gpt4 differ on different tasks? The author should make appropriate analysis .
3. In table 1, the result ‘83.0’ of column ‘Pure-time tasks-AMB.’ is not the best result. Suggest the authors to highlight the second results in the table as well.
4. In case study, there is a certain ambiguity in the first case (Math-CoT lacks prerequisites ‘1652 Jan 1st is Monday’). Suggest the authors to replace other case.

---

> ### Author Rebuttal · Authors · 2024-05-31
>
> Dear Reviewer QQVe,
>
> Thanks for the helpful comments! Hope the following responses address some of your questions:
>
> > Concern 1: Lack of analysis on comparison with other models in the ‘Main Results’ section.
>
> Thanks for your suggestion. We will add a detailed analysis of comparisons with LLMs for temporal reasoning, mathematical reasoning, and code generation in the next version.
>
> For example:
>
> **Comparison with LLMs for temporal reasoning:** TimeLLaMA is the only open-source model specially designed for making temporal predictions and generating time-related explanations. TIMO's performance is far better than TimeLLaMA (i.e., scoring 72.7 vs 38.6 on average), indicating that focusing solely on enhancing a single aspect of temporal capabilities does not lead to a comprehensive improvement.
>
> > Concern 2: Lack of comparison with GPT-4 in the ‘Ablation Analysis of Framework’ section.
>
> Thanks for your review.
>
> Actually, we have detailed the comparison between TIMO and GPT-4 in Section 6 "Performance comparison between TIMO and OpenAI GPT models".
>
> TIMO excels in pure-time tasks due to its ability to generate and select high-quality temporal preference pairs. However, TIMO still falls short in math-related tasks due to LLaMA's weaker mathematical reasoning compared to GPT-4.
>
> > Concern 3: In table 1, the result ‘83.0’ is not the best result.
>
> Sorry for the typo. In the revised Table 1, the best results will be bold, and the second results will be underlined.
>
> > Concern 4: In case study, there is a certain ambiguity in the first case (Math-CoT lacks prerequisites ‘1652 Jan 1st is Monday’). Suggest the authors to replace other case.
>
> Thanks for your suggestions! For a better demonstration, we will analyze the following case:
>
> ```markdown
> What is 13:01 + 14:43? (A) 5:50   (B) 0:43   (C) 3:44   (D) 4:48
> ```
>
> > Question1: Conducting experiments on other LLMs to verify TIMO's effectiveness.
>
> As shown in the table below, we conduct experiments on LLaMA3-8B to verify TIMO's effectiveness.
>
> Compared to vanilla LLaMA3-8B, TIMO shows an average improvement of **5.1** scores, with 1.2 scores in math-related tasks and 9 scores in time-related tasks.
>
> These results are consistent with the experiments conducted on LLaMA2-7B in our paper,  indicating TIMO's capability to enhance temporal reasoning abilities of different LLMs.
> | | Math-related task | Time-related task | Average  |
> | - | - | - | - |
> |LLaMA3-8B|81.4|79.6|80.5|
> |TIMO (LLaMA3-8B)| **82.6**| **88.6** | **85.6** |

---

> > ### Author Response · Authors · 2024-06-04
> > **Looking forward to further discussion**
> >
> > Dear Reviewer QQVe,
> >
> > To address your concerns, we have provided additional model comparisons, improved case studies, and conducted new experiments on LLaMA2-8B to verify the effectiveness of TIMO.
> >
> > With the rebuttal period closing soon, could you please check the response to see whether it mitigates your concerns?
> >
> > If you have any further inquiries or suggestions, please do not hesitate to reach out to us!
> >
> > Best,
> >
> > Authors of TIMO

---

### Official Review · Reviewer_Q4nZ · 2024-05-13

**Rating:** 7
**Confidence:** 5
**Ethics Flag:** 1

**Summary:**

This paper proposes a model, Timo, achieving new SOTA at 7B and 13B scales, which leverages the mathematical datasets to set a solid foundation model for temporal reasoning and enhance with self-critic temporal optimization method.

**Reasons To Accept:**

1. For temporal reasoning, it leverages the mathematical dataset as the foundational SFT data and uses DPO with temporal reasoning datasets for further improvement.
2. Experimental results demonstrate the performances of Timo are even better than GPT-4 in some areas. The analysis of the framework is insightful, and can deeply inspire other reasoning tasks.

**Reasons To Reject:**

1. More generalization of the framework should be validated in other reasoning tasks.

---

> ### Author Rebuttal · Authors · 2024-05-31
>
> Dear Q4nZ,
>
> Thanks for your feedback and insightful suggestions!
>
> Since temporal reasoning is an integrated task that requires arithmetic, logical, and commonsense reasoning abilities [1], our designed self-critic framework effectively handles these aspects and achieves significant improvements in temporal reasoning.
>
> Therefore, we believe our framework may also enhance other reasoning tasks that rely on the above three kinds of abilities. We will continue to explore the generalization of our framework across more reasoning tasks in the future.
>
> [1] TimeBench: A Comprehensive Evaluation of Temporal Reasoning Abilities in Large Language Models, arxiv-23

---

> > ### Author Response · Authors · 2024-06-04
> > **Looking forward to further discussion**
> >
> > Dear Reviewer Q4nZ,
> >
> > Thanks for your review! Considering the response deadline is approaching, we hope to have more discussions with you.
> >
> > If you have any further inquiries or suggestions, please do not hesitate to reach out to us!
> >
> > Best,
> >
> > Authors of TIMO

---

> ### Comment · Reviewer_Q4nZ · 2024-06-06
>
> Thanks for the response. I have no questions about the rebuttal and the further exploration of the generalization of the framework in more temporal reasoning tasks should be added in the revision. And I will keep the score unchanged.

---

### Decision · Program_Chairs · 2024-07-10

**Decision:**

Accept

**Comment:**

The reviewers agree that this paper has value and would contribute to the program in the conference. There were some concerns by one reviewer about not having experiments on held-out temporal reasoning datasets. The authors provided a proper rebuttal with new experiments that made the reviewer reconsider their position.